# Discrimination of Lung Cancer and Benign Lung Diseases Using BALF Exosome DNA Methylation Profile

**DOI:** 10.3390/cancers16152765

**Published:** 2024-08-05

**Authors:** Chinbayar Batochir, In Ae Kim, Eun Ji Jo, Eun-Bi Kim, Hee Joung Kim, Jae Young Hur, Do Won Kim, Hee Kyung Park, Kye Young Lee

**Affiliations:** 1Seasun Biomaterials, Inc., Daejeon 34015, Republic of Korea; chinba@seasunbio.com (C.B.); ejjo@seasunbio.com (E.J.J.); ebkim97@seasunbio.com (E.-B.K.); dowonkim@seasunbio.com (D.W.K.); hkpark@seasunbio.com (H.K.P.); 2Precision Medicine Lung Cancer Center, Konkuk University Medical Center, Seoul 05030, Republic of Korea; 20180618@kuh.ac.kr (I.A.K.); hjkim@kuh.ac.kr (H.J.K.); 20160475@kuh.ac.kr (J.Y.H.)

**Keywords:** lung cancer, bronchoalveolar lavage fluid (BALF), extracellular vesicles, methylation

## Abstract

**Simple Summary:**

Lung cancer diagnosis often requires invasive procedures due to the lack of effective early detection methods, particularly the insufficient specificity of current screening approaches. This study aimed to develop and validate epigenetic biomarkers from bronchoalveolar lavage fluid (BALF) exosome specimens capable of discriminating between lung cancer and benign lung diseases suspected of lung malignancy. Our findings indicate that combinations of epigenetic biomarkers derived from BALF exosomes can effectively support the discrimination of these clinical conditions with high specificity.

**Abstract:**

Benign lung diseases are common and often do not require specific treatment, but they pose challenges in the distinguishing of them from lung cancer during low-dose computed tomography (LDCT). This study presents a comprehensive methylation analysis using real-time PCR for minimally invasive diagnoses of lung cancer via employing BALF exosome DNA. A panel of seven epigenetic biomarkers was identified, exhibiting specific methylation patterns in lung cancer BALF exosome DNA. This panel achieved an area under the curve (AUC) of 0.97, with sensitivity and specificity rates of 88.24% and 97.14%, respectively. Each biomarker showed significantly higher mean methylation levels (MMLs) in both non-small cell lung cancer (NSCLC) and small cell lung cancer (SCLC) compared to non-cancer groups, with fold changes from 1.7 to 13.36. The MMLs of the biomarkers were found to be moderately elevated with increasing patient age and smoking history, regardless of sex. A strong correlation was found between the MMLs and NSCLC stage progression, with detection sensitivities of 79% for early stages and 92% for advanced stages. In the validation cohort, the model demonstrated an AUC of 0.95, with 94% sensitivity and specificity. Sensitivity for early-stage NSCLC detection improved from 88.00% to 92.00% when smoking history was included as an additional risk factor.

## 1. Introduction

Lung cancer continues to pose a significant health challenge globally, remaining a leading cause of cancer-related mortality. According to the World Health Organization, lung cancer accounts for approximately 2.2 million new cases and 1.8 million deaths annually [1,2,3]. The high mortality rate is largely attributed to the fact that lung cancer is often diagnosed at advanced stages, resulting in poor prognosis [4,5]. Primary lung cancer is divided into two main groups, NSCLC and SCLC, which differ in occurrence rates, aggressiveness, prognosis, and treatment strategies [6]. Approximately 85% of newly diagnosed lung cancer cases are NSCLC, while SCLC accounts for about 15%.

SCLC typically arises in the bronchi (central airways) and is characterized by its aggressive nature, with a short doubling time and a high rate of cell division. Symptoms often appear suddenly, typically within 10 weeks before diagnosis, and depend on the location and size of the primary tumor. The symptoms include coughing, wheezing, and hemoptysis (coughing up blood). Approximately 60% of SCLC patients are diagnosed with metastatic disease, with common metastasis sites including the liver, brain, adrenal glands, bones, and bone marrow [7]. Due to its rapid progression and aggressiveness, the 5-year survival rate for SCLC ranges from 4% for distant stages to 16% for localized stages. In contrast, NSCLC shows relatively higher 5-year survival rates, ranging from 10% to 82% depending on the stage progression [8]. Although NSCLC generally has a better prognosis, and is less aggressive than SCLC, the detection of both lung cancer types at early stages is crucial for improving survival rates. 

Although LDCT screening has shown promise in identifying peripheral pulmonary lesions, including lung nodules, and has been increasingly employed in national lung cancer screening programs in developed countries targeting lung cancer high-risk populations, its ability to differentiate between malignant and benign lesions is limited. This often leads to unnecessary invasive procedures, overdiagnoses, incidental findings, increased patient distress, and—although rare—radiation-induced cancers [9,10]. It is noteworthy that a substantial proportion of patients undergoing LDCT exhibit suspicious lung masses or solid nodules; yet, the diagnostic yield for identifying lung cancer from these findings is notably low, with less than 1% of such nodules being ultimately diagnosed as lung cancer [11]. Conversely, false negatives can result in missed opportunities for early cancer treatment [12]. Consequently, there is a critical need for less-invasive, accurate, and reliable methods for the discrimination of lung malignancies and benign lung diseases.

A promising approach for improving lung cancer diagnosis lies in the utilization of BALF, a readily accessible biofluid that offers a potential source of biomarkers. The procurement of BALF involves a minimally invasive procedure that can be performed repeatedly with minimal risk to the patient, thus providing a reliable medium for biomarker assessment in lung cancer diagnostics [13,14,15]. BALF encompasses cellular and non-cellular components from the bronchial and alveolar spaces, making it an excellent representation of the tumor microenvironment. This proximity to neoplastic tissues enhances its sensitivity and specificity for biomarker detection, especially in cases of locally advanced, non-metastasized lung cancer. Compared to other biofluids such as blood plasma, pleural fluid, and urine, BALF may offer superior diagnostic potential due to its direct interface with the pulmonary tumor. The diagnostic utility of BALF is further underscored by its ability to reflect localized pathological changes within the lung. By harvesting cellular contents and soluble factors from the alveolar and bronchial regions, BALF provides a concentrated source of tumor-derived biomarkers. This enhances the detection capabilities for early-stage malignancies and aids in the identification of the molecular signatures associated with lung cancer progression [16].

Epigenetic alterations, particularly DNA methylation patterns, have emerged as promising biomarkers for various cancers, including lung cancer [17,18]. Recent studies have shown that DNA methylation patterns from BALF could effectively discriminate lung cancer. For example, Li et al. reported that a methylation analysis of eleven genes that were specifically methylated in lung malignant tissues exhibited a 70% sensitivity and 82% specificity in distinguishing lung malignancy from benign conditions using BALF DNA [19]. Furthermore, Huiling et al. demonstrated that a combined analysis of *RASSF1A* and *SHOX2* methylation using BALF samples collected from patients with lung cancer and benign lung diseases showed 88.24% sensitivity and 81.25% specificity [20]. However, the discrimination of lung malignancy at an early stage remains challenging due to the limited sensitivity and specificity caused by the presence of non-tumor-derived DNA fractions, which could constitute the majority of BALF specimens [21]. 

Efforts in detecting lung cancer using liquid biopsy (blood) have been ongoing for decades, with several assays, such as PulmoSeek (AnchorDx, Fremont, CA, USA), Epi proLung BL Reflex Assay (Epigenomics AG, Heidelberg, Germany), and DELFI (Delfi Diagnostics, Baltimore, MD, USA), having been commercialized. Although liquid biopsy-based approaches offer numerous advantages, including minimal invasiveness, the ability to monitor tumor development, and the potential to represent tumor heterogeneity, they also have critical limitations. Specifically, their sensitivity and specificity are often insufficient, particularly in the early stages of cancer, due to the low concentration of tumor-derived biomolecules in the blood [22]. To overcome these challenges, we utilized BALF exosomes, which contain a high concentration of pure lung cancer-derived DNA, thereby potentially enhancing detection sensitivity and specificity compared to other liquid biopsies. This approach aims to achieve an accuracy comparable to standard tissue-based methods, as demonstrated in our previous studies. In these studies, we analyzed the most common *EGFR* mutations (exon 19 deletion and L858R) from the matched plasma, tissue, and BALF exosomes of 110 lung cancer patients using the same method. The concordance rate between tissue and plasma was determined to be 63.6%, while a concordance rate of 99.1% was identified between the BALF and tissue. Notably, the false positive rates for plasma and BALF were 13.6% and 0%, respectively, while the false negative rates for these samples were 51.5% and 1.5% [23,24].

To address these limitations and effectively discriminate lung malignancies at early stages from benign lung diseases, we conducted a screening of lung cancer-specific BALF exosome DNA biomarkers from a set of differentially methylated regions (DMRs) located in the major tumor suppressor genes previously reported to be linked to early cancer development. Exosomes, a type of extracellular vesicle (EV) ranging in size from 30 to 150 nanometers, are formed by the direct budding of the plasma membrane through endocytic pathways. They play a significant role in intercellular communication, transporting proteins, lipids, and genetic materials (DNA and RNA) [25,26]. Exosomes, which are found in various body fluids including urine, blood, bronchoalveolar lavage fluid (BALF), and cerebrospinal fluid, are increasingly recognized as valuable sources of cancer biomarkers. They also offer significant insights into the tumor microenvironment in cancer patients [27,28]. Given this, analyzing the biomolecules contained in exosomes offers a unique opportunity to explore cancer-specific biomarkers, as well as DNA methylation patterns, which contain direct cancer-derived biomolecules that are crucial for cell-to-cell communication, including those involved in cancer progression and metastasis [29]. 

In this study, we present a comprehensive methylation analysis utilizing real-time PCR to identify lung cancer-specific epigenetic biomarkers using BALF exosome specimens to address diagnostic challenges by leveraging the epigenetic insights from DNA methylation patterns in the BALF samples obtained from individuals suspected of having lung malignancies. Further validation and clinical investigations are warranted to fully establish the clinical utility of BALF-based testing and its integration into routine clinical practice.

## 2. Materials and Methods

### 2.1. Patients and Sample Collection

A total of 238 residual BALF samples were collected from patients diagnosed with either lung cancer or benign lung diseases, including non-malignant nodules, chronic obstructive pulmonary disease (COPD), pneumonia, and idiopathic pulmonary fibrosis (IPF), as well as those who were suspected of lung malignancies via routine clinical examinations, including pulmonary function test, chest X-ray, and chest tomography (CT) scans, conducted between 2020 and 2022. The cohort did not include samples from patients suspected of SARS-CoV-2 infection.

The BALF collection followed a standard protocol established in our hospital. Briefly, BALF was obtained from the sub-segmental bronchus at the site of the tumor mass during bronchoscopy. Approximately 50–70 mL of sterile isotonic saline was instilled, and a minimum of 10 mL of BALF was aspirated with the bronchoscope wedged at the tumor-containing segment. The BALF samples were allocated for routine cytological examination and microbial analysis before being utilized for methylation assays.

The handling and analysis of clinical samples were conducted in accordance with the approved protocol by the Institutional Review Board (KUMC 2020-10-009) of Konkuk University Medical Center. The research was conducted using anonymized leftover clinical samples following the principles outlined in the Declaration of Helsinki.

### 2.2. Exosome Isolation and DNA Extraction

Exosomes were purified from 2 mL of the BALF samples within 2 h of collection following the standard exosome purification method described in our previous studies [23,24]. Briefly, cells and debris, including cell contents, were removed through centrifugation at 1000× *g* for 10 min at 4 °C. Subsequently, the cell-free BALF specimen underwent ultracentrifugation at 200,000× *g* for 1 h at 4 °C using a Beckman Coulter ultracentrifuge (Beckman Coulter, Brea, CA, USA). The supernatant containing larger extracellular vesicles was carefully aspirated and discarded. The pellet containing exosome isolates was suspended in 200 μL of a 1X phosphate-buffered saline solution and stored at –80 °C until DNA extraction. 

Exosome DNA was purified using an Exo-HI DNA extraction kit (Exosignal, Seoul, Republic of Korea), which comprises a lysis buffer containing 10 mM Tris-HCl and 20% Triton X-100 to efficiently lyse the exosome membranes, following the manufacturer’s instructions. The purified DNA isolates were subsequently stored at −20 °C until further analysis.

### 2.3. DNA Methylation Testing

Marker screening was performed using an Epi-TOP™ MPP/Tumor Suppressor Assay (Seasun Biomaterials, Daejeon, Republic of Korea), which evaluates the methylation levels of 62 tumor suppressor genes known to exhibit differential methylation patterns in solid tumors, including lung cancer [30,31]. No standard cutoff value exists to classify the methylated and unmethylated loci in DNA methylation analyses. To achieve the highest clinical accuracy, the highest percent methylation ratio (PMR) value across the non-cancer clinical BALF samples, which aids in effectively discriminating between cancer and non-cancer samples with the highest AUC, was used as the cutoff threshold for each gene. Genes exhibiting PMR values above these cutoff thresholds were considered hypermethylated, while those with PMR values below the cutoff were considered non-methylated. Based on these criteria, the cutoff values to classify hypermethylated and unmethylated loci were determined as follows: 5.66 for *HOXA9*, 24.35 for *HOXD3*, 0.33 for *PCDH17*, 1.89 for *NID2*, 2.48 for *NPTX2*, 5.46 for *SFRP2*, and 0.91 for *RASSF1A*.

### 2.4. Statistical Analysis and Data Visualization

Statistical analysis and data visualizations were conducted using GraphPad Prism 10 software (GraphPad Software, San Diego, CA, USA). Sensitivity and specificity comparisons among groups were assessed using the Pearson chi-square test. All reported *p* values were two-sided, and a *p* value less than 0.05 was considered statistically significant. Receiver operating characteristic (ROC) curve analysis and calculation of the AUC were employed to evaluate the diagnostic efficacy of the seven DMRs individually and in combination as selected.

## 3. Results

### 3.1. Marker Selection

A methylation analysis of the tumor suppressor gene promoters, which have been previously reported to be linked to the development of solid cancers including lung adenocarcinoma, was performed using the DNAs extracted from the 138 exosome isolates purified from the clinical BALF specimens of patients suspected of lung malignancies identified via LDCT screening (Table 1).

The analysis utilized the Epi-TOP™ MPP/Tumor Suppressor assay, a real-time PCR test using methylation-specific restriction enzymes (Enzynomics, Daejeon, Republic of Korea), and methylation-specific peptide nucleic acid probes (Seasun Biomaterials, Daejeon, Republic of Korea) designed to specifically bind to methylated cytosine residues and to selectively amplify the methylated DNA copies following the cleavage of non-methylated copies with appropriate MSREs [30,31]. The PMR, representing the target methylation level, was determined by calculating the Ct value differences of each target gene and an internal control (human ACTB gene) as follows: PMR = 100/(1 + 2^Ct target − Ct control^) [32,33].

The thresholds for PMR values capable of accurately distinguishing BALF exosome DNA samples derived from non-cancer and cancer patients were established to achieve the highest clinical accuracy for each marker. Six DMRs located on *HOXA9*, *HOXD3*, *PCDH17*, *NID2*, *NPTX2*, and *SFRP2* genes exhibiting clinical accuracy over 75% irrespective of lung cancer subtype and cancer progression stages were selected as lung cancer-specific exosome DNA methylation biomarkers. Additionally, a DMR located on RASSF1A, showing the highest clinical specificity of 95.71% among the tested DMRs, was selected to augment the specificity of the marker combinations. However, it exhibited an adequate sensitivity rate (Table 2).

All selected biomarkers, except *RASSF1A*, demonstrated an area under the curve (AUC) of above 0.79 with statistically significant *p* values. *RASSF1A* exhibited an AUC of 0.64 when the *p* value between the methylation levels of non-cancer and cancer patient samples was 0.004, signifying a significant difference (Figure 1).

We cross-referenced the number of publications reporting the methylation of these selected DMRs with roles in lung cancer development or detection. For instance, Du et al. demonstrated that *HOXA9* and *RASSF1A* are hypermethylated in blood cfDNA samples from lung cancer patients. Their study revealed that a methylation panel incorporating these two genes could distinguish lung cancer from benign nodules with sensitivities and specificities of 86.7% and 81.4%, respectively [34]. Similarly, Cai et al. found that *HOXA9* methylation levels were negatively correlated with overall survival in lung cancer patients, suggesting its potential as a prognostic biomarker [35]. Han et al. reported that *HOXD3* is hypermethylated in patients with lung adenocarcinoma compared to normal controls, highlighting its utility in diagnosing lung adenocarcinoma [36]. Moreover, studies by Zhang et al. and Wang et al. showed that the hypermethylation of *SFRP2* and *NID2* significantly reduces their expression, promoting lung cancer development and invasion [37,38]. *PCDH17* downregulation and hypermethylation were identified in NSCLC clinical samples, suggesting its value as a diagnostic biomarker for NSCLC [39,40]. Although no publications explicitly link *NPTX2* methylation to lung cancer development, it has been reported that *NPTX2* functions as a tumor suppressor in various solid cancers, such as pancreatic cancer, prostate cancer, glioblastoma, and thymic epithelial tumors. This indicates its potential as a clinical tool for non-invasive cancer prognosis and monitoring disease progression [41,42,43].

### 3.2. Clinical Performance of the Selected Marker Combination

The AUC and clinical performance of the seven-gene-combined analysis for discriminating BALF samples collected from non-cancer and cancer patients were evaluated. The highest clinical performance was observed at a cutoff value where at least two of the seven genes exhibited PMR values exceeding the established PMR threshold for each marker. Using this cutoff (≥2), the seven-gene combination achieved an AUC of 0.97, with a clinical accuracy of 92.75% (Figure 1, Table 2). The clinical performance of the seven-gene combination across different lung cancer stages revealed a specificity of 97.14%, with sensitivities of 84.62% for the early-stage (I), 89.74% for advanced stages (II, III, and IV), and 89.47% for all NSCLC samples regardless of stage (Table 3). However, due to limited sample sizes, an individual analysis of the discrimination of NSCLC Stage II was inconclusive. These findings suggest that contributions from cancer-derived biomolecules and exosome release in BALF increase with cancer progression, which is consistent with previous reports [44,45,46].

### 3.3. Methylation Distributions of the Biomarkers

To assess the potential dependence of biomarker methylation levels among cancer and benign disease subtypes, we compared the PMR distributions and mean PMR (MPMR) values of each marker across non-cancer (nodule, pneumonia, tbc, COPD, and IPF) and cancer subtypes (adenocarcinoma, SqCC, SCLC, and LCLC). Each marker exhibited significantly higher MPMRs in all four cancer subtypes than in the control non-cancer groups (Appendix A). The MPMRs of each marker in non-cancer and cancer samples were confirmed to be significantly different, with fold changes ranging from 1.7 to 13.36 at statistically significant *p* values (Table 4). These findings indicate that the combined analysis of the methylation status of the seven genes could effectively serve its intended purpose, irrespective of the subtypes of lung cancer and benign conditions.

Furthermore, the methylation levels of each marker in relation to NSCLC stage progression were evaluated. A strong correlation was observed between the PMR values and NSCLC stage progression (Appendix A), with fold changes ranging from 1.33 to 6.39 between non-cancer and early NSCLC stages, as well as from 1.67 to 13.61 between non-cancer and advanced NSCLC stages, where all were at statistically significant *p* values (Table 5). These results support the notion that exosome release escalates with cancer progression. Moreover, the analysis of BALF exosome methylation demonstrates potential beyond screening and detection, extending to the monitoring of cancer progression.

### 3.4. Methylation and Patient Demographics 

Potential associations between the methylation patterns of the selected biomarkers and patient demographics, including sex (male/female), age (≥70/<70), and smoking history (smoker, including ex-smoker/never smoker) were examined within both non-cancer and cancer groups. In the non-cancer group, a modest increase in MPMR levels was observed with advancing patient age, irrespective of patient sex; however, this pattern did not reach statistical significance. Conversely, within the cancer patient group, no such age-related increase was evident. Instead, a moderate, albeit statistically insignificant, decrease in methylation levels was noted with increasing patient age (Table 6). This observation suggests that methylation of the biomarkers may induce a modest elevation with aging under normal conditions, whereas, in the context of cancer progression, methylation levels may undergo substantial augmentation, which is consistent with previous reports [47,48,49].

Significant or at least moderate elevations in MPMR values were observed across all seven markers depending on the smoking status of patients in both the cancer and non-cancer groups. Specifically, statistically significant MPMR elevations, with fold changes of 2.01, 1.36, and 2.61, were identified on the promoters of *HOXA9*, the *HOXD3* genes of cancer patients, and the *RASSF1A* promoter of non-cancer patients, respectively (Table 6). These findings suggest that the methylation levels of these genes may be influenced by the smoking status of patients. Therefore, a thorough validation, coupled with an assessment of the patient’s smoking history, is imperative before the clinical implementation of these markers.

### 3.5. Marker Validation

The clinical performance and discrimination power of the seven-gene-combined analysis were verified using an independent set of 100 additional BALF exosome-derived DNA samples collected from 65 lung cancer and 35 non-cancer patients with benign lung diseases who were suspected of lung malignancies through LDCT screening. Consistent with the discovery study, samples showing higher methylation levels in at least two out of seven markers above the PMR threshold values were considered positive for lung cancer. The seven-gene-combined analysis demonstrated an AUC of 0.95, with clinical sensitivity and specificity of 93.85% and 94.29%, respectively, in discriminating the samples derived from patients with lung cancer. 

An additional analysis, incorporating smoking history as a lung cancer risk factor, was performed in combination with the seven-gene-combined analysis. The combined analysis achieved an AUC of 0.96 at a statistically significant *p* value, with a sensitivity of 95.38%. Notably, the sensitivity for discriminating early-stage (I) NSCLC patients was increased from 88.00% to 92.00% when considering smoking history alongside the seven methylation markers (Figure 2, Table 7).

## 4. Discussion

In this study, we demonstrated the potential of BALF exosome DNA methylation biomarkers as a promising tool for the early detection and discrimination of lung cancer from benign conditions, addressing the limitations posed by a classical lung cancer screening method, i.e., LDCT. Our findings underscore the importance of leveraging epigenetic alterations, particularly DNA methylation patterns, to enhance the accuracy and reliability of lung cancer diagnosis, especially in the context of less invasive screening methods.

The comprehensive methylation analysis conducted in this study revealed a panel of seven epigenetic biomarkers that exhibit high sensitivity and specificity for discriminating lung cancer from benign conditions. The combination of these biomarkers demonstrated a remarkable AUC of 0.97, with sensitivity and specificity rates of 88.24% and 97.14%, respectively. Moreover, the clinical performance of this biomarker panel was consistent across different stages of NSCLC, highlighting its potential utility in stratifying patients based on disease severity. 

In addition to the selected DMR combinations that showed the highest diagnostic performance, individual DMRs located in other genes crucial for cell cycle regulation, DNA repair, and cellular functions such as *CDKN2A*, *DAPK*, *MLH1*, and *Septin9*, which were previously reported as hypermethylated in lung cancer samples [50,51,52], showed high efficacy in distinguishing lung cancer from benign conditions during the marker screening study. However, these genes were eliminated during the preliminary marker combined analysis as they did not enhance the clinical performance of the marker combinations despite their high individual performances.

With the widespread adoption of LDCT for early lung cancer screening, it is anticipated that patients with peripheral pulmonary lesions, such as pulmonary nodules, will continue to increase. While histological confirmation through invasive tissue sampling is typically performed when lung cancer is suspected, it may be challenging or risky to perform lung biopsy in cases where the lesion is small, difficult to target, or in cases of ground–glass opacity nodules, cavitary or cystic nodules, and consolidation-type lesions [53,54,55,56]. Practically, in such scenarios, it is common to initially observe the course and proceed with surgical biopsy by video-assisted thoracic surgery under general anesthesia if the possibility of lung cancer has been increased. Consequently, the majority of lung cancer surgeries are based on radiological findings without pre-operative tumor evaluation, potentially leading to unnecessary surgeries in patients with benign pulmonary diseases or causing diagnostic delays that contributes to higher recurrence rates, especially in NSCLC patients [57,58]. 

Given this context, there is a critical need for subsequent clinical studies to validate the utilization of BALF exosome DNA methylation profiles, which could potentially enhance the diagnostic rate of early-stage lung cancer and avoid unnecessary invasive tissue sampling or potential diagnostic delay. Particularly in East Asian countries, where there is a high frequency of peripheral-type lung cancer, adenocarcinoma, and lung cancer in females and never smokers, integrating our approach with LDCT screening could contribute to improving the therapeutic outcomes of early lung cancer patients.

It is essential to acknowledge the limitations of our study, particularly regarding the validation of biomarkers in patients with SCLC. Due to the limited availability of samples and the relatively low prevalence of SCLC compared to NSCLC, the validation of the biomarkers in SCLC patients was constrained. Future studies with larger cohorts of SCLC patients are warranted to validate the efficacy and reliability of the BALF exosome DNA methylation biomarkers in this subset of lung cancer. Additionally, the correlation analysis of the methylation status of the selected genes and their mRNA or protein expression, which would provide fundamental insights supporting the regulatory pathways of those genes, could not be performed due to the limited availability of sample volume. We plan to conduct this analysis in our further studies. 

Furthermore, while our study demonstrates promising results regarding the feasibility and efficacy of BALF exosome biomarkers for lung cancer diagnosis, there are practical considerations that need to be addressed for the integration of this assay into routine clinical practice. Factors such as sample collection protocols, assay standardization, and scalability need to be carefully evaluated to ensure the feasibility and reproducibility of the assay in real-world clinical settings. Additionally, evaluating the cost-effectiveness and integration of BALF exosome DNA methylation testing with existing diagnostic modalities will be essential to determine its economic impact and clinical utility.

## 5. Conclusions

This study underscores the potential of BALF exosome DNA methylation biomarkers in addressing the limitations of LDCT screening, facilitating the differentiation between lung nodules and lung cancer. By exploiting the epigenetic landscape, this approach offers a promising avenue for refining diagnostic precision and advancing personalized treatment strategies. Our findings suggest that BALF exosomes contain pure cancer-specific biomolecules, making it a potent biomaterial for cancer detection, particularly at early stages. By addressing the limitations of current diagnostic modalities and considering the practical aspects of assay implementation, BALF exosome biomarkers hold significant promise for improving the clinical management of lung nodules and advancing precision medicine in lung cancer care. Comprehensive validation and further clinical investigations are imperative to fully establish the clinical utility of this pioneering methodology in managing lung nodules.

## Figures and Tables

**Figure 1 cancers-16-02765-f001:**
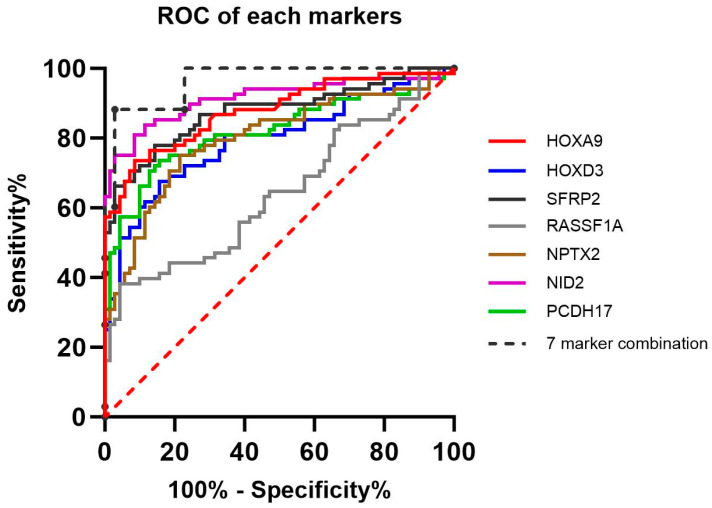
The ROC of each gene and seven-gene-combined analysis.

**Figure 2 cancers-16-02765-f002:**
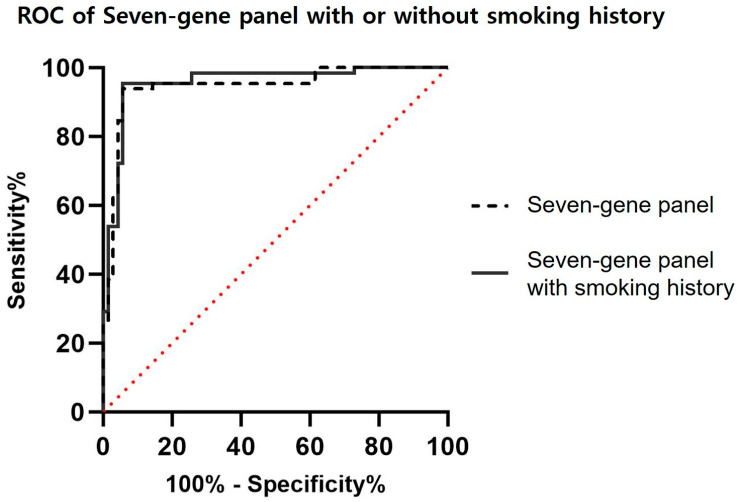
ROC of the seven-gene-combined analysis calculated with or without smoking history.

**Table 1 cancers-16-02765-t001:** Patient characteristics.

		Discovery (n, %)	Validation (n, %)
Age (mean)	Non-cancer	68.4	68.5
	Cancer	72.1	70
Sex			
Non-cancer	Male	45 (64%)	18 (51%)
	Female	25 (36%)	17 (49%)
Cancer	Male	55 (81%)	36 (55%)
	Female	13 (19%)	29 (45%)
Smoking status			
Non-cancer	Never	33 (47%)	24 (69%)
	Minimal ex-smoker	2 (3%)	-
	Ex-smoker	6 (9%)	4 (11%)
	Smoker	21 (30%)	5 (14%)
	NA ^1^	8 (11%)	2 (6%)
Cancer	Never	19 (28%)	29 (45%)
	Minimal ex-smoker	1 (1%)	3 (5%)
	Ex-smoker	3 (4%)	6 (9%)
	Smoker	45 (66%)	27 (42%)
Histology			
Non-cancer	Nodule	18 (26%)	6 (17%)
	Ground–glass nodules (GGN)	-	9 (26%)
	Pneumonia	19 (27%)	6 (17%)
	Tuberculosis (Tbc)	19 (27%)	9 (26%)
	COPD	11 (16%)	5 (14%)
	IPF	1 (1%)	-
	ND ^2^	2 (3%)	-
Cancer	Adenocarcinoma	25 (37%)	50 (77%)
	Squamous cell carcinoma (SqCC)	30 (44%)	10 (15%)
	SCLC	10 (15%)	2 (3%)
	LCLC	2 (3%)	-
	Not otherwise specified (NOS)	1 (1%)	3 (5%)
Cancer stage	NSCLC I	13 (19%)	25 (38%)
	NSCLC II	1 (1%)	3 (5%)
	NSCLC III	20 (29%)	10 (15%)
	NSCLC IV	18 (26%)	25 (38%)
	NSCLC relapsed	3 (4%)	1 (2%)
	NSCLC N/A	2 (9%)	-
	SCLC LD	2 (3%)	-
	SCLC ED	9 (13%)	1 (2%)

^1^ Not available; ^2^ not determined.

**Table 2 cancers-16-02765-t002:** Clinical performance of each marker.

Gene	Sensitivity (95% CI)	Specificity (95% CI)	Accuracy (95% CI)	AUC (95% CI) *p* < 0.05
*HOXA9*	73.53% (61.43–83.50)	91.43% (82.27–96.79)	82.61% (75.24–88.53)	0.88 (0.82–0.94)
*HOXD3*	80.88% (69.53–89.41)	65.71% (53.40–76.65)	77.97 (64.99–80.37)	0.79 (0.72–0.87)
*PCDH17*	73.53% (61.43–83.50)	84.29% (73.62–91.89)	78.99 (71.23–85.45)	0.82 (0.74–0.89)
*NID2*	83.82% (72.90–91.64)	88.57% (78.72–94.93)	86.23 (79.34–91.50)	0.91 (0.86–0.97)
*NPTX2*	75.00% (63.02–87.71)	77.14% (65.55–86.33)	76.09 (68.09–82.93)	0.80 (0.73–0.88)
*RASSF1A*	38.24% (26.71–50.82)	95.71% (87.98–99.11)	67.39 (58.90–75.12)	0.64 (0.55–0.73)
*SFRP2*	77.94% (66.24–87.10)	85.71% (75.29–92.93)	81.88 (74.43–87.92)	0.87 (0.81–0.93)
Seven-gene-combined analysis	88.24% (78.13–94.78)	97.14% (90.06–99.65)	92.75 (87.08–96.47)	0.97 (0.94–0.99)

**Table 3 cancers-16-02765-t003:** Clinical performance of the seven-gene-combined analysis across NSCLC stages.

Gene	NSCLC Stage	Sensitivity (95% CI)	Specificity (95% CI)	AUC (95% CI) *p* < 0.05
Seven-gene-combined analysis	I	84.62% (57.77–97.27%)	97.14% (90.17–99.49)	0.87 (0.76–1.00)
	III	90.00% (69.90–98.22%)		0.93 (0.84–1.00)
	IV	94.44% (74.24–99.72%)		0.95 (0.88–1.00)
	II/III/IV	89.74% (76.42–95.94%)		0.93 (0.86–0.99)
	I–IV	88.46% (77.03–94.60%)		0.92 (0.86–0.98)
	I–IV + N/A ^1^	88.89% (77.81–94.81%)		0.92 (0.86–0.98)
	I–IV + N/A + RE ^2^	89.47% (78.88–95.09%)		0.92 (0.87–0.98)

^1^ Stage unknown; ^2^ relapsed.

**Table 4 cancers-16-02765-t004:** Mean PMR values of each marker.

Gene	Mean PMR		Fold Change ^1^ *p* < 0.05
	Non-Cancer (n = 70)	Cancer (n = 68)	
*HOXA9*	2.505	25.04	9.996
*HOXD3*	20.80	35.42	1.703
*PCDH17*	0.272	3.634	13.360
*NID2*	0.901	6.437	7.144
*NPTX2*	2.255	13.79	6.115
*RASSF1A*	0.403	1.992	4.943
*SFRP2*	3.279	12.67	3.864

^1^ Fold change = cancer MPMR/non-cancer MPMR.

**Table 5 cancers-16-02765-t005:** The MPMR values and the fold changes in each marker between the cancer and non-cancer groups.

Gene	MPMR Non-Cancer	NSCLC Stage I		NSCLC Stage II/III/IV	
		MPMR	Fold Change ^1^	MPMR	Fold Change
*HOXA9*	2.505	10.16	4.06	26.72	10.67
*HOXD3*	20.8	27.75	1.33	34.70	1.67
*PCDH17*	0.2722	1.74	6.39	3.71	13.61
*NID2*	0.9007	4.13	4.59	6.81	7.56
*NPTX2*	2.255	6.31	2.80	14.77	6.55
*RASSF1A*	0.4033	1.21	3.01	1.27	3.16
*SFRP2*	3.279	9.30	2.84	12.98	3.96

^1^ Fold change = cancer MPMR/non-cancer MPMR.

**Table 6 cancers-16-02765-t006:** Clinical performance of each marker based on patient’s sex, age and smoking history.

Gene	Sex				Age				Smoking			
	Non-Cancer		Cancer		Non-Cancer		Cancer		Non-Cancer		Cancer	
	Fold Change ^1^	*p* Value	Fold Change ^1^	*p* Value	Fold Change ^2^	*p* Value	Fold Change ^2^	*p* value	Fold Change ^3^	*p* Value	Fold Change ^3^	*p* Value
*HOXA9*	1.11	0.689	1.49	0.173	1.07	0.790	0.93	0.755	1.50	0.109	2.01	0.013
*HOXD3*	0.95	0.617	1.31	0.071	1.13	0.261	0.93	0.503	1.25	0.062	1.36	0.012
*PCDH17*	0.84	0.726	1.24	0.658	1.61	0.325	0.70	0.292	1.27	0.636	1.43	0.383
*NID2*	0.88	0.531	1.11	0.768	1.09	0.647	1.07	0.793	1.31	0.207	1.30	0.390
*NPTX2*	1.34	0.416	1.78	0.244	1.30	0.423	1.37	0.333	1.51	0.221	1.58	0.235
*RASSF1A*	1.55	0.304	4.44	0.106	1.38	0.396	0.46	0.065	2.62	0.022	2.70	0.105
*SFRP2*	0.94	0.720	0.93	0.794	1.04	0.797	0.79	0.268	1.23	0.201	1.21	0.440

^1^ Fold change = MPMR(male)/MPMR(female); ^2^ fold change = MPMR(age ≥ 70)/MPMR(age < 70); ^3^ and fold change = MPMR(smoker including ex-smoker)/MPMR(never smoker).

**Table 7 cancers-16-02765-t007:** Clinical performance of seven-gene-combined analysis.

Method	Test Group (Cancer Stage)	Sensitivity (95% CI)	Specificity (95% CI)	Accuracy (95% CI)	AUC (95% CI) *p* < 0.05
Seven-gene panel	Total sample	93.85% (84.99–98.30)	94.29% (80.84–99.30)	94.00% (87.40–97.77)	0.95 (0.90–0.99)
	Stage I	88.00% (70.04–95.83)		91.67% (81.61–97.24)	0.92 (0.83–0.99)
	Stage II/III/IV	97.37% (86.19–99.93)		95.89% (88.46–99.14)	0.96 (0.92–1.00)
Seven-gene panel + smoking history	Total sample	95.38% (87.10–99.04)		95.00% (88.72–98.36)	0.96 (0.91–1.00)
	Stage I	92.00% (73.97–99.02)		93.33% (83.82–98.15)	0.96 (0.91–1.00)
	Stage II/III/IV	97.37% (86.19–99.93)		95.89% (88.46–99.14)	0.96 (0.92–1.00)

## Data Availability

The data presented in this study are available from the corresponding authors upon reasonable request.

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
