# Peer review of "Discrimination of Lung Cancer and Benign Lung Diseases Using BALF Exosome DNA Methylation Profile"

_cancers, 2024, doi:10.3390/cancers16152765_

Round 1
Reviewer 1 Report
Comments and Suggestions for Authors
Discrimination of lung cancer and benign lung diseases using BALF exosome DNA methylation profile
1. In the abstract section, I would suggest authors to start abstract with mentioning lung cancer and benign lung disease. Then you should mention briefly why this study is important. Further, you strat with method. This is important to make clear story, that should be visible in abstract section.
2. Wherever you mentioned NSCLC and SCLC for the first time, it is necessary to write their full names along with the abbreviations.
3. In the introduction section,
……..Please mention the epidemiology of the disease, that will will help to understand the current prevalence of the disease..
….it is important to discuss briefly about NSCLC and SCLC, also include how they differ from each other, their respective occurrence rates, primary causes, and main symptoms.
……It is very important to highlight about the exosomes in detail for their origin and size, how it differs to microvesicles of other extracellular vesicles. I would suggest authors to cite these papers https://www.mdpi.com/2227-9059/9/10/1373 , https://www.mdpi.com/2072-6694/16/12/2179
4. Method section…
…….Author should discuss the methods that were used previously to distinguish lung cancer and benign lung diseases, as well as their limitations.
……Before start any experiment with exosomes, after isolation of exosomes, it is important to confirm it by western blotting using specific markers such CD63, TSG, CD91 and electron microscopy
5. In lines 60-62, it is mentioned that "Recently, numerous studies have shown that DNA methylation pattern from BALF can effectively discriminate lung cancer." It would be appropriate to discuss some studies here.
6. In Patients and Sample Collection section, author should include a detailed breakdown of the number of samples from each specific condition (lung cancer, non-malignant nodules, COPD, pneumonia, and IPF) to understand the distribution better.
7. The sample collection period of this study is from 2020 to 2022. Were any patients affected by the COVID-19 pandemic during this period? If yes, then it could affect the health of their lungs and potentially affect the results of the study. Clarify this issue.
Comments on the Quality of English LanguageExtensive english editing is required.
Author Response
Comments 1: In the abstract section, I would suggest authors to start abstract with mentioning lung cancer and benign lung disease. Then you should mention briefly why this study is important. Further, you start with method. This is important to make clear story, that should be visible in abstract section.
Response 1: Thank you for your suggestion. We have revised the abstract to begin with mentioning lung cancer and benign lung diseases, emphasizing the importance of discriminating between them and addressing the challenges involved. Please refer to lines 16-18 of the abstract for these updates.
Comments 2: Wherever you mentioned NSCLC and SCLC for the first time, it is necessary to write their full names along with the abbreviations.
Response 2: Thank you for your comment. We have revised the abstract accordingly to include the full names along with the abbreviations NSCLC (Non-Small Cell Lung Cancer) and SCLC (Small Cell Lung Cancer). Please refer to lines 23-24 of the abstract for these corrections.
3. In the introduction section,
Comments 3-1: Please mention the epidemiology of the disease, that will help to understand the current prevalence of the disease.
Response 3-1: Thank you for your suggestion. We have included a brief discussion on the epidemiology of lung cancer, covering aspects such as prevalence, mortality rates, and factors contributing to its high mortality. Please refer to lines 35-41 of the Introduction for this addition
Comments 3-2: it is important to discuss briefly about NSCLC and SCLC, also include how they differ from each other, their respective occurrence rates, primary causes, and main symptoms.
Response 3-2: We have revised the Introduction as per your request, incorporating a discussion on NSCLC and SCLC, highlighting their differences, occurrence rates, mortality, and the importance of early detection. Please refer to lines 38-55 of the Introduction for this information.
Comments 3-3: It is very important to highlight about the exosomes in detail for their origin and size, how it differs to microvesicles of other extracellular vesicles. I would suggest authors to cite these papers https://www.mdpi.com/2227-9059/9/10/1373 , https://www.mdpi.com/2072-6694/16/12/2179
Comments 3-3: Thank you for your suggestion. We have revised the Introduction to include a brief section on exosomes, providing general information about their origin, size, and their role in cancer development and diagnostics. We appreciate the references you shared, and we have cited them in the manuscript. Please refer to lines 116-126 of the Introduction for these updates.
4. Method section…
Comments 4-1: Author should discuss the methods that were used previously to distinguish lung cancer and benign lung diseases, as well as their limitations.
Response 4-1: We have incorporated a section in the Introduction discussing current methods used to distinguish lung cancer from benign lung diseases, along with their respective advantages and limitations. Please refer to lines 94-111 of the Introduction for this addition.
Comments 4-2: Before start any experiment with exosomes, after isolation of exosomes, it is important to confirm it by western blotting using specific markers such CD63, TSG, CD91 and electron microscopy
Response 4-2: Thank you for your consideration. In this study, we utilized the same established method for isolating exosomes from clinical BALF samples and extracting exosome DNA as in our previous studies. Our lab has extensive experience in exosome isolation and the use of BALF exosome samples. Previously, we confirmed the purity of exosome isolates and the presence of DNA within these isolates using cryogenic and immune electron microscopy, which provide direct and precise visualization of exosomes. Additionally, we verified the size distribution of purified exosomes using Nanosight, a nanoparticle tracking analysis. Given our solid foundation in exosome purification and DNA isolation, our current study focused more deeply on exosome DNA testing. For your reference, attached below are our previous publications on exosome research:
1. Hur JY et al., Extracellular vesicle-derived DNA for performing EGFR genotyping of NSCLC patients. Mol Cancer. 2018 Jan 27;17(1):15.
2. Hur JY et al., Extracellular vesicle-based EGFR genotyping in bronchoalveolar lavage fluid from treatment-naive non-small cell lung cancer patients. Transl Lung Cancer Res. 2019 Dec;8(6):1051-1060
3. Kim HJ et al., Liquid biopsy using extracellular vesicle-derived DNA in lung adenocarcinoma. J Pathol Transl Med.2020 Nov;54(6):453-461.
4. Lee SE et al., Genomic profiling of extracellular vesicle-derived DNA from bronchoalveolar lavage fluid of patients with lung adenocarcinoma. Transl Lung Cancer Res. 2021 Jan; 10(1): 104–116
5. Kim IA et al., Extracellular Vesicle-Based Bronchoalveolar Lavage Fluid Liquid Biopsy for EGFR Mutation Testing in Advanced Non-Squamous NSCLC. Cancers (Basel). 2022 May 31;14(11):2744
6. Hur JY et al., Characteristics and Clinical Application of Extracellular Vesicle-Derived DNA. Cancers (Basel). 2021 Jul 29;13(15):3827.
We hope this information adequately addresses your query.
Comments 5: In lines 60-62, it is mentioned that "Recently, numerous studies have shown that DNA methylation pattern from BALF can effectively discriminate lung cancer." It would be appropriate to discuss some studies here.
Response 5: We have incorporated examples of studies that utilize BALF DNA methylation to discriminate lung cancer, as you suggested. Please refer to lines 82-93 of the Introduction for these additions.
Comments 6: In Patients and Sample Collection section, author should include a detailed breakdown of the number of samples from each specific condition (lung cancer, non-malignant nodules, COPD, pneumonia, and IPF) to understand the distribution better.
Response 6: Details of the clinical samples are described in Table 1 (page #5).
Comments 7: The sample collection period of this study is from 2020 to 2022. Were any patients affected by the COVID-19 pandemic during this period? If yes, then it could affect the health of their lungs and potentially affect the results of the study. Clarify this issue.
Response 7: We added a statement that the cohort did not include samples from patients suspected of COVID-19. Please see materials and methods Line 141-142.
Comments on the Quality of English Language
Extensive English editing is required.
Response: We appreciate this advice. We have substantially revised the text with the help of a native speaker

Reviewer 2 Report
Comments and Suggestions for Authors
The manuscript "Discrimination of lung cancer and benign lung diseases using 2 BALF exosome DNA methylation profile" is an intriguing study.
However, the study contains severe flaws:
The authors mentioned BALF exosomes in their manuscript. However, the protocol for isolating exosomes from BALF samples, including the supplementary material, is not stated anywhere in the publication.
Another issue in the report is the characterization of the exosomes. How did the authors characterize the exosomes?
Furthermore, exosomes were not screened for particular markers indicating their endocytic origin.
The work is excellent, but no researcher would proceed with the investigation unless the isolated vesicles were confirmed and characterized first.
Author Response
Comments 1: The authors mentioned BALF exosomes in their manuscript. However, the protocol for isolating exosomes from BALF samples, including the supplementary material, is not stated anywhere in the publication.
Response 1: We apologize for the confusion regarding the protocol for exosome isolation. The protocol for exosome isolation from BALF and subsequent DNA extraction from purified exosome isolates is detailed in the Materials and Methods section (Lines 154-165). Additionally, to clarify this, we have revised the title of section 2.2 to "Exosome Isolation and DNA Extraction" (Line 153).
Comments 2: Another issue in the report is the characterization of the exosomes. How did the authors characterize the exosomes? Furthermore, exosomes were not screened for particular markers indicating their endocytic origin. The work is excellent, but no researcher would proceed with the investigation unless the isolated vesicles were confirmed and characterized first.
Response 2: Thank you for your consideration. In this study, we utilized the same established method for isolating exosomes from clinical BALF samples and extracting exosome DNA as in our previous studies. Our lab has extensive experience in exosome isolation and the use of BALF exosome samples. Previously, we confirmed the purity of exosome isolates and the presence of DNA within these isolates using cryogenic and immune electron microscopy, which provide direct and precise visualization of exosomes. Additionally, we verified the size distribution of purified exosomes using Nanosight, a nanoparticle tracking analysis. Given our solid foundation in exosome purification and DNA isolation, our current study focused more deeply on exosome DNA testing. For your reference, attached below are our previous publications on exosome research:
1. Hur JY et al., Extracellular vesicle-derived DNA for performing EGFR genotyping of NSCLC patients. Mol Cancer. 2018 Jan 27;17(1):15.
2. Hur JY et al., Extracellular vesicle-based EGFR genotyping in bronchoalveolar lavage fluid from treatment-naive non-small cell lung cancer patients. Transl Lung Cancer Res. 2019 Dec;8(6):1051-1060
3. Kim HJ et al., Liquid biopsy using extracellular vesicle-derived DNA in lung adenocarcinoma. J Pathol Transl Med.2020 Nov;54(6):453-461.
4. Lee SE et al., Genomic profiling of extracellular vesicle-derived DNA from bronchoalveolar lavage fluid of patients with lung adenocarcinoma. Transl Lung Cancer Res. 2021 Jan; 10(1): 104–116
5. Kim IA et al., Extracellular Vesicle-Based Bronchoalveolar Lavage Fluid Liquid Biopsy for EGFR Mutation Testing in Advanced Non-Squamous NSCLC. Cancers (Basel). 2022 May 31;14(11):2744
6. Hur JY et al., Characteristics and Clinical Application of Extracellular Vesicle-Derived DNA. Cancers (Basel). 2021 Jul 29;13(15):3827.
We hope this information adequately addresses your query.

Reviewer 3 Report
Comments and Suggestions for Authors
The manuscript by Batochir et al. presents a methylation analysis employing real-time PCR to develop a less invasive diagnostic tool for lung cancer using BALF exosomes.
The authors highlight the importance of use of the BALF as a potential source of biomarkers for lung cancer diagnosis and allows the harvesting of cellular contents of the bronchial and alveolar space, serving as an excellent biological material of the tumor microenvironment.
However, the data provided raises some important questions.
By assessing DNA methylation authors write
- line 182 Each marker showed significantly higher MPMRs in all four tested cancer subtypes than in control non-cancer groups.
Methylation, a key epigenetic modification, plays a significant role in lung cancer by influencing gene expression without altering the DNA sequence. In general, hypermethylation of gene promoters results in gene silencing and loss of protein function.
According to the authors' research, the function of the gene should be silenced or reduced and cannot serve as a risk factor for the development of cancer in this case according to methylation paradigm. It appears that these genes are expressed more strongly in non-cancer groups than in cancer groups.
For example, activation of HOXA9 is negatively prognostic marker in malignant glioblastoma (GBM) patients. In colon cancer (CRC), HOXA9 expression is elevated in tissues compared to the normal epithelium, HOXA9 might drive CRC development and growth through regulating stem cell function.
Studies have shown that the human HOXD3 overexpression was found to regulate cell adhesion in human erythroleukemia HEL cells, induce coordinate metastasis-related gene expression, and enhance the motility and invasiveness of human lung cancer A549.
In the case of promoter methylation, the authors show, gene expression should be reduced, if at all. This is in contrast to other studies that have shown increased cancer-specific expression of HOXD3.
- Thus, correlation analysis of the effect of DNA methylation requires gene expression at the mRNA and protein levels.
- Please provide these data for all genes studied.
- Are there any comparative data between exosomal and cellular contexts?
- There is a need to clarify key genes associated with methylation in lung cancer (CDKN2A, MGMT, MLH1, KRAS, EGFR, etc.).
- Why are none of them, except RASSF1A, present among the methylated genes known to be methylated in lung cancer?
Comments on the Quality of English LanguageMinor editing required.
Author Response
Comments 1: line 182 Each marker showed significantly higher MPMRs in all four tested cancer subtypes than in control non-cancer groups.
Methylation, a key epigenetic modification, plays a significant role in lung cancer by influencing gene expression without altering the DNA sequence. In general, hypermethylation of gene promoters results in gene silencing and loss of protein function.
According to the authors' research, the function of the gene should be silenced or reduced and cannot serve as a risk factor for the development of cancer in this case according to methylation paradigm. It appears that these genes are expressed more strongly in non-cancer groups than in cancer groups.
For example, activation of HOXA9 is negatively prognostic marker in malignant glioblastoma (GBM) patients. In colon cancer (CRC), HOXA9 expression is elevated in tissues compared to the normal epithelium, HOXA9 might drive CRC development and growth through regulating stem cell function.
Studies have shown that the human HOXD3 overexpression was found to regulate cell adhesion in human erythroleukemia HEL cells, induce coordinate metastasis-related gene expression, and enhance the motility and invasiveness of human lung cancer A549.
In the case of promoter methylation, the authors show, gene expression should be reduced, if at all. This is in contrast to other studies that have shown increased cancer-specific expression of HOXD3.
- Thus, correlation analysis of the effect of DNA methylation requires gene expression at the mRNA and protein levels. Please provide these data for all genes studied.
Response 1: We completely agree with your comment. However, due to the limited volume of clinical samples (as we used leftover BALF specimens after routine microbiological testing using the majority of the initially collected specimens), conducting the correlation analysis you suggested is challenging.
We have included this statement in the discussion section as a limitation of our study. Please see the discussion section, lines 372-376. While we could not provide the correlation data you suggested, we have added detailed descriptions of the methylation markers selected in this study in the results section. Independent studies have shown that all the selected genes have been previously reported to be methylated in lung cancer or in other types of solid cancers (in the case of NPTX2), and they could be potential biomarkers for cancer detection and representatives of the tumor microenvironment. Since these genes have already been reported to be methylated in lung cancer samples in other publications, we believe that the correlation study of mRNA expression and methylation is not immediately crucial, allowing us to focus on the methylation studies. Please refer to lines 222-240 of the Results section for this addition.
We appreciate your understanding and valuable feedback. Also, we have been planning to conduct these correlation studies in our future research to further support the fundamental role of the selected genes.
Comments 2: Are there any comparative data between exosomal and cellular contexts?
Response 2: Unfortunately, we do not have any comparative data between exosomal and cellular contents. We have been focused on exosome isolation and the analysis of mutations or methylation in exosome DNA for approximately 20 years. Based on our extensive experience, we have found that BALF cellular contents primarily consist of normal lung epithelial cells and do not adequately represent tumor characteristics. Therefore, we have concentrated our efforts on exosome DNA, which has shown more promise in reflecting the molecular profile of lung tumors.
Comments 3: There is a need to clarify key genes associated with methylation in lung cancer (CDKN2A, MGMT, MLH1, KRAS, EGFR, etc.).
Response 3: Thank you for your valuable suggestion. In our marker screening, we included key methylation genes previously reported to play regulatory roles in cancer development, such as CDKN2A, MLH1. For this study, our primary focus was on identifying the optimal combination of genes that would provide the highest discrimination performance between lung malignancies and benign lung diseases. Although some of these key genes demonstrated adequate sensitivities to lung cancer samples, they did not enhance the AUC of the combined analysis significantly. Therefore, we selected genes that showed the highest performance for inclusion in our panel. For the clarification, we have added a brief discussion about this in the Discussion, lines 339-346. Thank you again for your insightful feedback.
Comments 4: Why are none of them, except RASSF1A, present among the methylated genes known to be methylated in lung cancer?
Response 4: We appreciate your question and believe the previous discussions partially address this concern. While our initial test set included several key genes known to be methylated in lung cancer, such as CDKN2A and MLH1, our focus was on identifying the optimal combination of genes that provide the highest discrimination performance between lung malignancies and benign lung diseases. Although these key genes demonstrated adequate sensitivity to lung cancer samples, they did not significantly enhance the AUC of the combined analysis. As a result, we selected the genes that showed the highest performance, which included RASSF1A. This approach allowed us to create a more effective panel for our specific study objectives. Thank you for your understanding.
Comments on the Quality of English Language: Minor editing required
Response: We appreciate this advice. We have substantially revised the text with the help of a native speaker

Round 2
Reviewer 1 Report
Comments and Suggestions for Authors
The authors addressed all my concerns, therefore, I recommend it to accept for publication in the journal.
Reviewer 3 Report
Comments and Suggestions for Authors
The authors were generally responsive to the comments of the reviewers. The revised manuscript has been significantly improved. Correlation analysis of the methylation status of selected genes and their mRNA or protein expression will hopefully provide evidence to support the significance of these genes in cancer development in future studies.